# Elevated Temperature Baseplate Effect on Microstructure, Mechanical Properties, and Thermal Stress Evaluation by Numerical Simulation for Austenite Stainless Steel 316L Fabricated by Directed Energy Deposition

**DOI:** 10.3390/ma15124165

**Published:** 2022-06-12

**Authors:** Abhilash Kiran, Ying Li, Martina Koukolíková, Michal Brázda, Josef Hodek, Miroslav Urbánek, Ján Džugan, Srinivasan Raghavan, Josef Odehnal

**Affiliations:** 1COMTES FHT a.s., Prumyslova 995, 334 41 Dobrany, Czech Republic; abhilash.rub@gmail.com (A.K.); martina.koukolikova@comtesfht.cz (M.K.); michal.brazda@comtesfht.cz (M.B.); josef.hodek@comtesfht.cz (J.H.); miroslav.urbanek@comtesfht.cz (M.U.); jan.dzugan@comtesfht.cz (J.D.); 2Makino Asia Pte Ltd., 2 Gul Ave, Singapore 629649, Singapore; srinivasanr@makino.com.sg; 3Department of Material Science and Technology, University of West Bohemia, Univerzitni 2732/8, 301 00 Pilsen, Czech Republic; odehnal@kmm.zcu.cz

**Keywords:** directed energy deposition, austenite stainless steel 316L, baseplate heating, numerical modeling, thermal stress, tensile properties

## Abstract

In the present study, the effect of material deposition at the elevated temperature baseplate on the microstructure and mechanical properties was investigated and correlated to the unique thermal history by using numerical simulation. Numerical results agreed well with the experimental results of microstructure and mechanical properties. Numerical results revealed a significant decrease in temperature gradient and a 40% decrease in thermal stress due to material deposition on the elevated temperature baseplate. The reduced thermal stress and temperature gradient resulted in coarser grain features, which in turn led to a decrease in hardness and tensile strength, especially for the bottom region near the baseplate. Meanwhile, no significant effect could be found for ductility. In addition, an elevated temperature baseplate promoted less heterogeneity in hardness and tensile properties along the building direction. The current work demonstrates a collective and direct understanding of the baseplate preheating effect on thermal stress, microstructure and mechanical properties and their correlations, which is believed beneficial for the better utilization of baseplate preheating positive effects.

## 1. Introduction

Directed Energy Deposition (DED) is one of the metal deposition techniques in the Additive Manufacturing (AM) process. The DED process utilizes a focused heat source (laser or electron beam) to melt feedstock material in the form of powder or wire to build a three-dimensional structure [1,2]. Localized heating by a focused heat source creates melt pools on the target surface. Unlike in Powder Bed Fusion (PBF,) where materials are spread in a tray, DED employs a technique to supply materials directly to the melt pool. This creates a unique application window, such as less support deposition with a high production rate, deposition of functionally graded materials and a unique advantage in the field of repairing worn-out parts, surface treatment, porous coatings, etc. [3,4].

The wide application of material deposition with a focused heat source has been hindered due to several hurdles. Employment of a concentric thermal heat source and complicated thermal cycles during the deposition process results in residual stress. It leads to the distortion of the deposited part and the baseplate [5,6]. Shim et al. investigated the crack evolution on DED- and PBF-processed M4 steel. They found out that the inconvenience in material deposition on the baseplate at room temperature in the deposition of M4 steel results in significant cracks and inadequate fusion at the deposited layer-substrate interface [7]. Similarly, the deposition of M2 high-speed steel caused a failure due to crack development in the deposited structure in PBF [8]. Functionally graded material deposition using DED also is prone to crack development [9].

A common effective solution for all the above difficulties is preheating the baseplate. Respective researchers successfully proved the effectiveness of preheating to address the corresponding challenges. Lu et al. [6] achieved significant success in reducing residual stresses and final distortion by 80.2% and 90.1%, respectively. Shim et al. [7] reported baseplate preheating with an induction heater in the metal deposition of the HSS M4 powder, beneficial in reducing the crack formation. It was shown that coarser columnar grains and larger secondary dendritic arm spacing were generated due to the lower solidification rate in the baseplate preheating case compared to equiaxial finer grains produced without preheating. Similarly, Kiran et al. [10] found an increase in average grain size for increased baseplate temperatures during the single track deposition of DED austenite 316L steel. Wei et al. [9] observed that laser synchronous preheating was an effective means of improving deposition and crack suppression in the laser deposition for Inconel625/Ti6Al4V graded material. Jendrzejewski et al. [11] studied the preheating effect of a stellite SF6 coating on the X10Cr13 chromium steel using numerical simulation. Numerical calculation of preheating revealed that the stress state was below the material tensile strength during a preheating condition at 500 °C. Microstructural characterization corresponded to numerical calculation. A decrease in the number of cracks was observed in the preheated sample compared to that conducted at room temperature.

In the last few years, extensive valuable outcomes have contributed to our understanding of the correlations of the process parameters, microstructure and mechanical properties of DED-fabricated components [12,13,14,15]. However, most of those available studies discuss deposition at room temperature. It is well known that baseplate preheating can effectively reduce residual stress, thermal distortion as well as cracking in compositionally gradient materials fabricated via DED [16,17,18]. Therefore, it is of great importance to study systematically the baseplate preheating effect on the microstructure and mechanical properties in order to exploit the utilization of baseplate preheating during the manufacturing of AM parts.

Extensive previous reports mentioned above have contributed to our understanding about the baseplate preheating effect on the reduction of residual stress and distortion generated during AM. However, these works focused on either the reduction of the negative effects of concentrated heat during the deposition process by using numerical simulations in [19,20], or the relationships between the baseplate preheating effect and microstructure and mechanical properties, without providing thermal information. A collective and direct understanding of thermal history-induced microstructure and mechanical properties for AM materials is needed in order to have a clearer picture of the baseplate preheating effect for better utilization of its positive effects.

Therefore, the aim of the present work is to provide a collective understanding of the baseplate preheating effect and thermal history by using numerical simulation–microstructure–mechanical properties correlations. In addition, different from the baseplates preheated before deposition in previous reports, the baseplate is maintained at an elevated temperature until the end of deposition. This elevated temperature deposition was implemented to reduce the difference in temperature gradients along the deposition direction, which may help to achieve better consistency in the cooling rate from the layers near the baseplate until the end of deposition. In this condition, slow cooling was achieved after the end of deposition. Three regions were considered along the building direction to evaluate the mechanical and microstructural results. A numerical model was built for thermal and mechanical simulation. A thermal model was built along with two heat sources to simulate the heating condition. Numerical results were validated with the respective experimental data. Numerical results were used to evaluate the thermal history and thermal stress along the built direction. The present numerical simulation of deposition at elevated temperatures was compared with that at room temperature. The results for the thermo-mechanical simulation of baseplates at room temperature were derived from our previous work [20].

## 2. Materials and Methods

### 2.1. Materials

Austenitic stainless steel 316L was used as the material for the powder feedstock and baseplate. Austenitic stainless steel 316L does not undergo phase transformation in the solid state. Austenitic stainless steel 316L (Sandvik Osprey LTD, Neath, UK) powder with a particle size range of 50–150 µm was used. A baseplate with dimensions of 95 × 95 mm^2^ and thickness of 6 mm was used. To reduce the influence of the residual stress, the baseplate was heat-treated at 400 °C for four hours.

### 2.2. Experimental Setup

The material was deposited by using the InssTek MX-600 metallic deposition system (InssTek, Daejeon, Korea). The DED process parameters provided by the DED machine manufacturer were used for material deposition [20].

The heating setup was designed for the InssTek MX-600 metallic deposition system. A thermally insulated heating system was designed as per the baseplate holder setup available in the InssTek MX-600 metallic deposition system. The heating unit is shown in Figure 1a. The baseplate was fixed using bolts from the external support structure, as shown in Figure 1b. Fiber glass insulation was built around the heating coil and baseplate to minimize the heat loss. Insulation also helped to control the desired temperature in the baseplate throughout the process.

### 2.3. In Situ Temperature Measurement

The temperature was measured in situ on the top surface of the baseplate using a thermocouple of type “K”. The desired temperature on the top surface was maintained via thermocouple reading using the heating system. The thermocouple (TC_Top) was welded at a distance of approximately 2 mm from the contour deposition track of the cube. The data from the TC_Top helped to control heat from the heating unit. The thermocouple position is marked in Figure 2. The demission of the baseplate and cuboid structure is also shown in Figure 2.

### 2.4. Tensile Test

The miniaturized tensile tests (MTT) were carried out to evaluate the tensile properties [21,22]. A half section of the deposited cuboid cut from centerline is shown in Figure 3a. Three locations were considered for tensile testing. Four samples were prepared for each investigated location. The first set was near the baseplate, the second set was at the middle of the height of the deposited structure, and the third set was at the top, as shown in Figure 3b. The orientation of the tensile samples is shown in Figure 3c. The testing specimens were extracted from the deposited material by a wire electric discharge machine (WEDM) in accordance with the geometry depicted in Figure 3d.

The quasi-static tensile tests were conducted at a strain rate of 0.00007 s^−1^ using a universal testing machine, the Zwick/Roll Z250, with a load capacity of 2.5 KN at room temperature. The testing procedure was performed according to the standards ISO 6892-1 (Metallic materials—Tensile testing—Method of test at room temperature). The deformation was measured with the use of an optical system based on the Digital Image Correlation method.

### 2.5. Hardness and Microstructural Evaluation Methods

Hardness profiles HV 10 (load 10 kg) were measured using a laboratory hardness tester, the Struers DuraScan 50 (EMCO-TEST Prüfmaschinen GmbH, Kuchl, Austria), equipped with ecos WorkflowTM software (EMCO-TEST Prüfmaschinen GmbH, Kuchl, Austria). The hardness measurement was carried out in accordance with ISO 6507-11: Vickers hardness measurement. The measurement was performed from the bottom surface of the baseplate with a step of 1 mm. The indentation profile of the hardness test is depicted in Figure 4. Standard metallographic preparation of hardness samples consisted of grinding followed by polishing (final steps—Nap 1 µm + OP-S (Colloidal silica—0.25 μm)), which was performed on a Tegramin 30 (Struers GmbH, Ballerup, Denmark). The microstructures were revealed by etching in V2A solution and photographed using a light microscope, the Nikon Eclipse MA200 (Nikon, Tokyo, Japan), equipped with the NIS Elements 5.2 digital image processing and analysis software (Nikon, Tokyo, Japan). The detailed microstructural observation and electron backscatter diffraction (EBSD) were performed on a scanning electron microscope, JEOL IT 500 HR (JEOL Ltd., Tokyo, Japan), with the EDAX Hikari Super camera (EDAX LLC, Mahwah, NJ, USA) at a step size of 2.5 µm, for an analyzed area of 1076 µm × 1080 µm, with an acceleration voltage of 30 kV, scanning speed of 100 diffractograms per second, and 5 × 5 binning. The EBSD maps were processed in 75× magnification. The data acquisition, analyses, and post-processing were performed using the software TEAM 4.5 (EDAX LLC, Mahwah, NJ, USA) and EDAX OIM Analysis™ Version 8.0 (EDAX LLC, Mahwah, NJ, USA).

### 2.6. Numerical Simulation

The deposition process at elevated temperature was simulated in two steps using uncoupled thermal and mechanical steps. A finite element model was developed for both thermal and mechanical simulation using 3DExperience software (Dassault Systèmes, Vélizy-Villacoublay, France). The melt pool created from the laser was simulated using a concentric heat source model [20]. The moving heat source was set to 500 W power as per the experimental input data.

#### 2.6.1. Thermal Analysis

A progressive material deposition strategy that represented the DED process of new material addition with time was implemented [20]. As the upper-deposited layers were exposed to ambient conditions, this resulted in continuous evolution of the cooling surface with time. Therefore, progressive cooling was applied to the surface of the deposited material. The detailed method and theories for thermal analysis used in the present work can be found in our previous work [20].

#### 2.6.2. Mechanical Analysis

The mechanical analysis was conducted using the temperature histories computed by the thermal analysis as the input data.

The mechanical equation is based on the equation of static equilibrium [23]:(1)(∇σ+F)=0
where ∇ is the divergence operator, σ is the stress tensor, and F is the body force.

As the material for baseplate and powder was austenitic stainless steel, which does not undergo solid-state phase transformation, the total strain rate is given as [24]:(2)$$\dot{\varepsilon }={\dot{\varepsilon }}^{e}+{\dot{\varepsilon }}^{p}+{\dot{\varepsilon }}^{th}$$
where ${\dot{\varepsilon }}^{e}$ is the elastic strain, ${\dot{\varepsilon }}^{p}$ is the plastic strain, and ${\dot{\varepsilon }}^{th}$ is the thermal strain. The elastic strain was modeled using the temperature-dependent Young’s modulus and Poison’s ratio according to isotropic Hook’s law. The plastic strain was computed using temperature-dependent mechanical properties, a linear kinematic hardening model, and kinematic hardening [25].

### 2.7. FEM Model

Meshed models used for both thermal and mechanical simulation had the same mesh size with respective boundary conditions. Linear heat transfer elements (DC3D8) were selected for the thermal simulation. Linear elements (C3D8) from the 3DExperience library were used for mechanical analysis. The FEM model for the 3D solid cuboid structure consisted of 31,360 elements (1.5 mm mesh size) and 36,225 nodes. The FEM model for the baseplate consisted of 9216 elements (2 mm mesh size) and 12,005 nodes, as depicted in Figure 5.

Two assumptions were considered to reduce the complexity in numerical model development for the present work. A uniform thermal heat loss from the entire baseplate surface was assumed. The emissivity parameter was kept constant at 0.1, in accordance with the literature [26]. The heat transfer coefficient by convection was optimized for deposition on the baseplate at the 600 °C condition. A higher convection rate of 47 W/m^2^ K was set compared to deposition on the baseplate at room temperature. The surrounding temperature was set to 27 °C.

The next assumption was implemented for the mechanical boundary condition. The bottom plane of the baseplate below the deposition area was restricted in all three directions.

## 3. Results

This section presents results on the comparison between deposition at an elevated temperature and at room temperature in terms of the effect on thermal history and thermal stress by numerical simulation, and the influence on microstructure and mechanical properties, respectively.

### 3.1. Thermal Analysis

The end of deposition and beginning of the cooling process is shown in Figure 6. Once deposited on the baseplate, the upper surface temperature of the material can reach up to 600 °C. The temperature on the surface at the end of deposition drops by only around 50 °C from the start of deposition, compared to around 250 °C during material deposition on the baseplate at room temperature [20]. Slow cooling was maintained with the help of the heating system. Cooling was prolonged to nine times more compared to without baseplate heating [20].

Two thermal heat sources were employed for thermal simulation. One thermal source represented the laser interacting with the material deposition. The movement of the laser was defined as per the laser movement data derived from the experiment. The other one was created as a stationary heat source within the baseplate. This was optimized over time as per the thermocouple results. It represents the external heat source setup in the experiment. The nodal temperature on the surface of the baseplate 2 mm away the contour of the deposition is plotted in Figure 2. The simulation results agree well with the thermocouple-measured data.

### 3.2. Mechanical Analysis

Mechanical simulation was conducted with the same meshed model used for thermal simulation with the respective structural boundary condition. Mechanical simulation results were compared with experimentally determined results (contour method) [1,27,28]. The contour method provides normal stress perpendicular to the cut surface (X-Z plane). Similarly, the numerical model was also cut in the X-Z plane, as shown in Figure 7. The normal stress to the X-Z plane (Y-component stress) from numerical mode was compared with the contour method results. For precise comparison, three paths were considered in the deposited structure, marked with dark lines, as depicted in Figure 7. The first path was located at a height of 3 mm from the top surface of the baseplate. This path corresponded to the 12th layer deposition. The middle path was at a height of 18.75 mm, where the 75th layer was deposited. The topmost path was located at 35.5 mm, corresponding to the 142nd layer deposition.

Y-component stress values for numerical results at three paths are depicted in Figure 8 with respect to contour method results. The blue-colored plot represents the results originating from the contour method, while the orange plot demonstrates the normal stress to the X-Z plane generated from numerical calculation. The contour results at the edge are considered inaccurate. It is likely due to error occurring during the cutting process. The cutting edge influences residual stress relaxation [20,29].

The results for the path 3 mm from the baseplate surface are depicted in Figure 8a. The results show a slight difference between the two measurements, which could be mainly caused by the structure boundary conditions applied to the baseplate below the deposition area. Moreover, the numerical results show negative Y-component stress in this path, similar to the contour method results. Meanwhile, results for the second path, at a height of 18.75 mm from the baseplate’s top surface, agree well with the experimental results, as can be seen in Figure 8b. The result of the third path, at a height of 35.5 mm, in Figure 8c, shows a shift in Y-component stress to the positive at a distance of approximately 5 mm in the experimental results, and a similar variation was reported for the numerical calculation. In the same manner, numerical results were also estimated, as shown in Figure 8c.

### 3.3. Temperature Profile and Thermal Stress Evolution

The nodal temperature and stress were extracted from the validated model, and the results can be seen in Figure 9. Nodal temperature represents the temperature evolution at a specific point, and it helps to understand the temperature profile during the deposition process inside the structure. Five nodes on different layers were considered for analysis. It can be seen from Figure 9a that the notable development of the baseplate preheating effect occurs in a narrow window of approximately 110 °C nodal peak temperature for the considered layers. The window of minimum and maximum temperature is depicted in Figure 9a. Meanwhile, a larger window of approximately 440 °C of the peak temperature was reported for deposition on the baseplate at room temperature [20]. This was due to rapid heat flow during deposition at the initial layers to the baseplate at room temperature. The elevated temperature of 600 °C deposition successfully minimizes the inconsistency. For simplicity, the narrow window of peak temperature can be assumed as homogeneous heat flow to the previous deposited layers. The rise in the peak temperature near the baseplate layer node in Figure 9a reveals that the elevated temperature of the baseplate supports the retention of heat for a comparatively longer time than that of deposition on the baseplate at room temperature. Interestingly, the peak temperature on the upper layers does not rise considerably when maintaining the baseplate at an elevated temperature of 600 °C. It only increases the peak temperature for the initial layers. In the case of the deposition on a baseplate at room temperature, it was observed that the peak temperature in the initial layers was considerably lower than the peak in the upper layers due to continuous heat source interaction with the deposited material [20]. From this evidence, elevated temperature baseplate deposition shows its limited influence at the initial few layers.

Extending the cooling phase using external heating on the baseplate seems to be less effective for the upper layers. It can be effective in avoiding a sudden drop near room temperature in a short time span. The baseplate heating helps to hold the temperature to 300 °C, whereas deposition on the baseplate without heating falls below 100 °C for the same duration. Cooling was prolonged from 300 °C onwards.

Results of peak nodal stress for the respective layers are shown in Figure 9b. It can be seen that thermal stress on the baseplate node and 12th layer reached higher values compared to upper layers. It can be noticed that the peak stress reached approximately 150 MPa from the 75th layer onwards. This indicates that the thermal peak stress reaches a constant magnitude after deposition at the initial few layers. Compared to deposition on the baseplate at room temperature, the peak stress of the considered layers significantly falls. The highest stress was found for the node on the baseplate of 350 MPa [20], whereas the highest stress of 200 MPa was reported for the same layer in the case of elevated temperature deposition. Elevated temperature baseplate deposition successfully reduced thermal stress development during deposition by around 43% on the baseplate. A drop of 37%, 40%, 35%, and 28% was noticed for the 12th, 75th, 142nd, and 160th layers, respectively. Reducing thermal stress during deposition and limiting the stress development within the respective material strength could avoid failure during the deposition process, which is crucial during the deposition of a composite material.

An important development can be noticed in the cooling phase. The start of the cooling phase is indicated in Figure 9b. Compared to the stress variation in the cooling phase without external heat and cooling in the ambient room temperature condition [20], here, thermal stress does not rise at the start of cooling. A sudden jump in the thermal stress could be observed due to the interruption of heat input to the structure from the laser source. Meanwhile, in the present case, slow cooling shows an impact on stress relaxation after deposition.

### 3.4. Microstructure Evaluation

The microstructure evolution for the deposits on the baseplate at room temperature and at 600 °C is compared in this section. Figure 10 shows the comparison of the microstructure near the baseplate for samples produced without baseplate preheating (WPH) and with elevated temperature baseplate preheating of 600 °C (EPH). It can be clearly seen that deposition on the elevated temperature baseplate can significantly influence the morphology and the size of grains in the vicinity of the baseplate by controlling the temperature gradient G and the local solidification growth rate R. For samples without baseplate preheating, large amounts of fine columnar grains and relatively smaller amounts of equiaxed grains growing along the heat flow direction are observed in the first deposited layer, depicted in Figure 10a. In contrast, for samples produced by elevated temperature baseplate preheating of 600 °C, smaller amounts of columnar grains but large amounts of coarser equiaxed grains are found in the first deposited layer due to the lower cooling rate and G/R ratio, as shown in Figure 10b.

The importance of the effect of baseplate preheating on the cooling rate and consequently the microstructure of the first few layers in the baseplate transition region was also noted by Moheimani et al. [30], who confirmed the formation of the different microstructures near the baseplate and along the building direction. Heat starts to accumulate in the deposit structure due to the repeated scanning cycles of the laser. The temperature rises during the deposition of each layer, increasing in the build direction in the case of material deposition on the baseplate at room temperature [20].

It is well known that heat transfer into the baseplate is decreased with the layer built up in AM, resulting in lower cooling rates for the upper-deposited layers. These variations result in inhomogeneous microstructures and mechanical properties in different regions of a part. Thereby, it may explain the slight increase in grain size in the second deposited layer for the WPH sample, as shown in Figure 10a. However, noticeably coarser grains begin to occur in the second deposited layer for the EPH sample in Figure 10b. This means that the elevated temperature preheating at 600 °C in the present study can significantly reduce the capability of heat extraction via the baseplate and the thermal gradients during the first deposited layer. The changes in grain shape, size, and orientation in the first layer between WPH and EPH samples are due to change in the heat flow rate to the baseplate. Similar results were reported by Kiran et al. [10], where a considerable change in average grain size and orientation was observed during single track deposition at 500 °C. At the second layer deposition, lower thermal gradients cannot allow heat generated to successfully conduct via the first-deposited layer. The heat flow could be reduced compared to the heat flow rate during deposition at the first layer to the baseplate. Thus, a coarse grain structure is generated for WPH parts compared to that of WPH due to a lower cooling rate.

In order to examine the preheating effect on the upper-deposited layers far from the baseplate, microstructure analysis for the EPH sample at the bottom, middle, and top regions was performed with comparison to the WPH sample. EBSD IPF maps for both EPH and WPH samples at three regions are depicted in Figure 11. It can be seen that EPH samples have relatively coarse grains at the three regions in Figure 11d–f compared to the WPH samples in Figure 11a–c, as indicated by larger amount of columnar grains with increased width (perpendicular to the building direction in the former case). Moreover, increased grain size is observed in higher deposited layers for both samples compared to the first deposited layers. As discussed above, grain size is expected to increase with the increase in build height due to the reduced cooling rate with increasing build height. However, no evident difference in grain size and crystallographic orientation could be found for both WPH and EPH samples at the bottom, middle, and top regions. Uneven columnar grains with a weak intensity of crystallographic orientation could be seen in both conditions, which may be caused by the location-dependent thermal stress and/or heterogeneous nucleation on the partially melted areas, such as the overlapped areas or the areas near the solidification front [31].

### 3.5. Mechanical Properties

In the following section, mechanical properties are compared between samples created without preheating and with elevated temperature preheating conditions.

#### 3.5.1. Hardness

Figure 12 depicts the hardness dependence of the distance from the bottom of the baseplate. It can be seen that the baseplate hardness was almost constant for both cases. The average value for the EPH sample (170 ± 4 HV/10) is smaller than that of the WPH sample (194 ± 7 HV/10), but higher than the conventional heat-treated baseplate (162 ± 7 HV/10). The decreased hardness for the EPH sample is probably due to the formation of the coarse grain features. In addition, compared to the WPH sample without preheating, a small scatter in the results is observed for the EPH sample. Within the scatter, no clear dependency of the results on the location can be found. If at all, the values are slightly lower at the middle region than the bottom and top regions for WPH samples. However, homogeneous values for hardness measurement could be obtained throughout the EPH bulk, which can be attributed to the homogeneous microstructure generated by the slow cooling during elevated temperature baseplate heating. A similar trend of the hardness measurement was also reported by Moheimani et al. [30].

#### 3.5.2. Tensile Properties

Figure 13 depicts the engineering stress–strain curves of EPH and WPH samples at different locations. It can be seen from Figure 13 that, similar to the hardness result, EPH samples exhibit a lower yield and ultimate strength but a higher elongation compared to WPH samples, due to the coarse grain characteristics formed in the elevated temperature baseplate condition. As stated in previous reports, the decreased yield strength is expected to cause better fracture toughness but a worse fatigue property [32,33]. However, the DED-manufactured SS316L produced without preheating and extended preheating still showed superior tensile strength to the conventionally processed cast and annealed counterparts [34].

In order to evaluate the preheating effect on the tensile properties with respect to sample location, the location dependence of 0.2% yield strength, ultimate strength, and elongation of WPH and EPH specimens is shown in Figure 14, and the summary results are given in Table 1.

It can be seen from Figure 14 that the value for the 0.2% yield and ultimate strength of WPH samples decreases with the increase in height along the build direction. This is due to the decreased cooling rate with the increase in height along the build direction, caused by the heat accumulation during the deposition process. This microstructure-induced mechanical property change with the location along the build direction is observed and discussed in previous reports [35,36,37]. However, this heterogeneity in microstructure and tensile properties is significantly alleviated for EPH samples. Almost homogenous tensile properties can be achieved for elevated temperature preheated specimens, as revealed by the similar value of 0.2% yield strength at the bottom, middle, and top regions of the EPH samples in Figure 14 and Table 1. Regarding the elevated temperature preheating effect range, there is a remarkable difference in specimens obtained from different locations. Though a striking decrease in tensile strength for EPH samples was observed at different locations relative to that of WPH samples, this reduced tensile strength decreases with the height along the build direction, i.e., the value of 0.2% yield strength of EPH samples decreases in a sequence of bottom region, middle region, and top region. This implies that the elevated temperature preheating primarily influences the bottom region, then the middle region, and has the least impact on the top region. This is understandable, since the heat at the bottom region mainly dissipates via the baseplate and is directly dependent on the thermal condition of the baseplate. In contrast, the heat at higher regions is transferred via the solidified, previously deposited lower layers and is less sensitive to the thermal condition of the baseplate. Thus, the preheating effect decreases gradually with the increase in deposit height. In addition, the heat conduction in the top region is more complicated than that at the bottom and middle regions, since the heat in the top region is mainly dissipated via radiation and convection to the surrounding atmosphere. Compared with a large difference in strength caused by the baseplate effect, no significant difference in ductility is observed for preheated samples at three different locations relative to samples without preheating. Only a slight increase in the middle region is shown due to the loss of strength in the middle region. In a summary, baseplate preheating is found to influence the microstructure and tensile properties throughout the deposited bulk in the current study, especially for the bottom region near the baseplate.

## 4. Conclusions

The article demonstrates the effect of material deposition on the elevated temperature baseplate preheating compared to room temperature baseplate heating. The following outcomes are noted for the three sections:A thermal model was simulated using two heat sources. Introducing a stationary secondary heat source helps to simulate material deposition with external heating. Both the thermal and mechanical results agree well with the experimental data. Further thermal boundary definition needs to be improved to increase the accuracy and stability of the numerical results. The thermal evaluation at different layers reveals an increase in the peak temperature near the baseplate due to elevated temperature baseplate heating. Consequently, a nodal peak stress drop was noticed in different layers inside the deposited structure. The promising results demonstrate the lower risk of failures such as crack development due to internal stress. These results promote the deposition of multiple materials.Elevated temperature baseplate preheating contributes to a coarse grain feature throughout the bulk, especially for the first deposited layer. Remarkably, coarse grains are stimulated to grow from the second deposited layer, in contrast with finer grains with a slower grain growth rate in the case of deposition on the room temperature baseplate. A more homogenous microstructure can be achieved due to the continuously elevated temperature baseplate heating.Elevated temperature baseplate preheating yields a significant decrease in hardness and yield strength throughout the deposit, especially for the bottom region near the baseplate. Meanwhile, no significant effect on strain can be found. In addition, less heterogeneity in hardness and yield strength measurement can be achieved.

## Figures and Tables

**Figure 1 materials-15-04165-f001:**
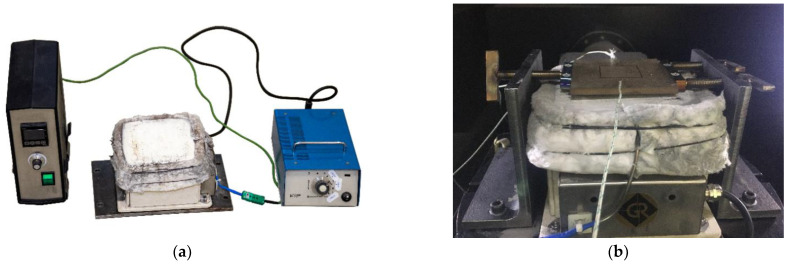
(**a**) Preheating setup in the deposition chamber. (**b**) Baseplate holding setup on the heating plate.

**Figure 2 materials-15-04165-f002:**
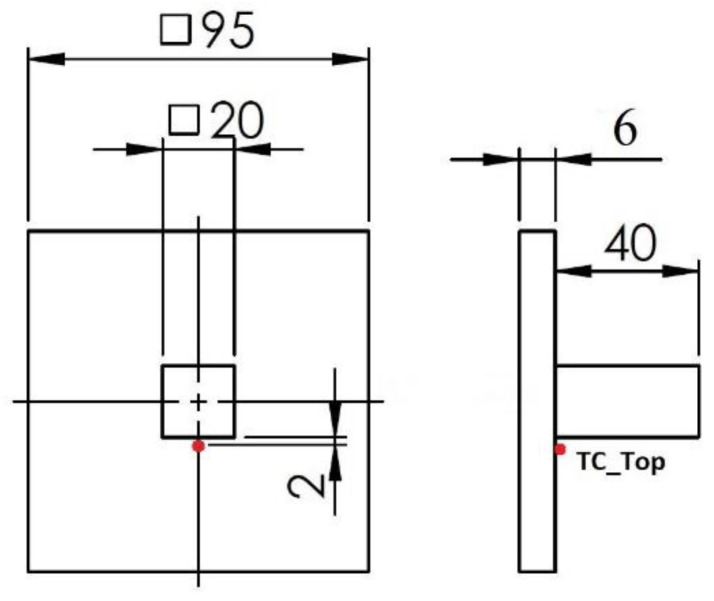
Thermocouple position and structure dimension.

**Figure 3 materials-15-04165-f003:**
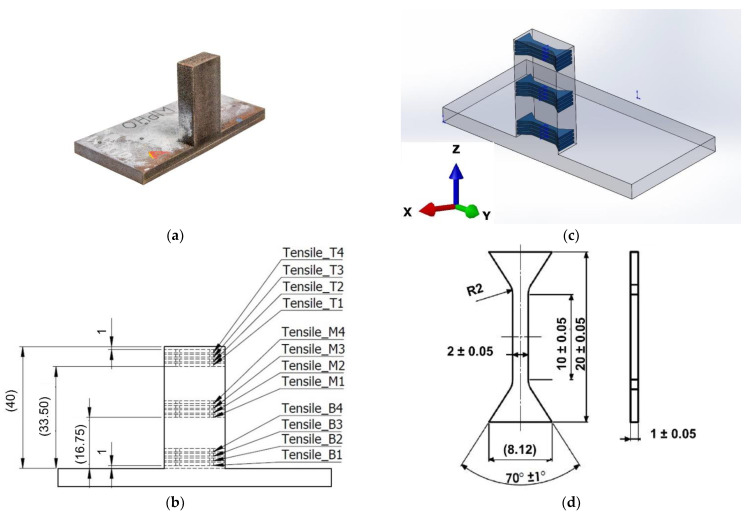
(**a**) Cut cuboid structure for tensile sample preparation. (**b**) Three section positions from the baseplate top surface. (**c**) Sample orientation in the deposited structure. (**d**) Tensile samples’ dimensions.

**Figure 4 materials-15-04165-f004:**
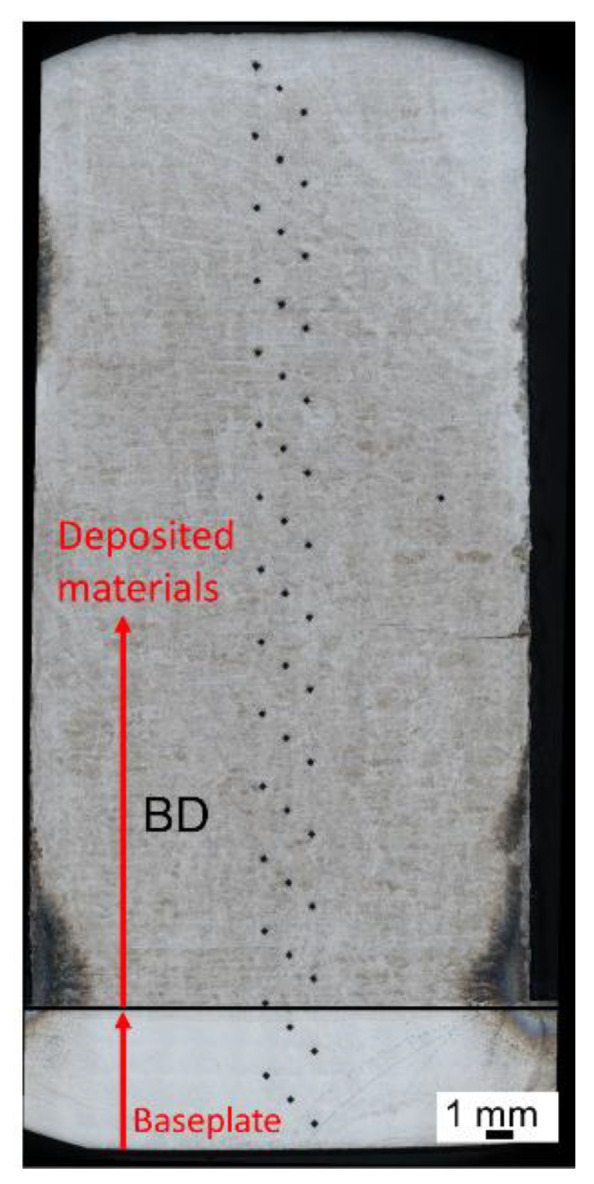
Indentation profile of hardness test. The indentation was performed along the building direction of deposition.

**Figure 5 materials-15-04165-f005:**
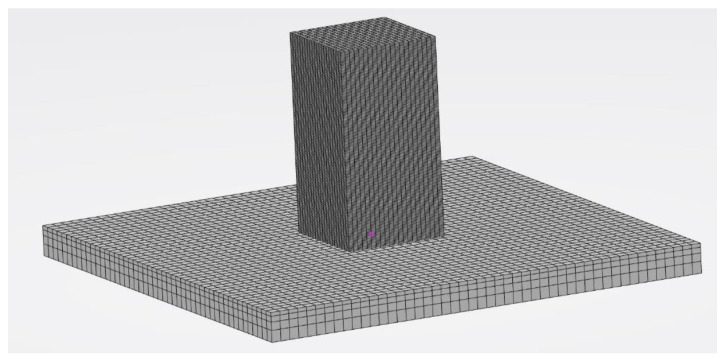
FEM meshed model for thermal and mechanical analysis.

**Figure 6 materials-15-04165-f006:**
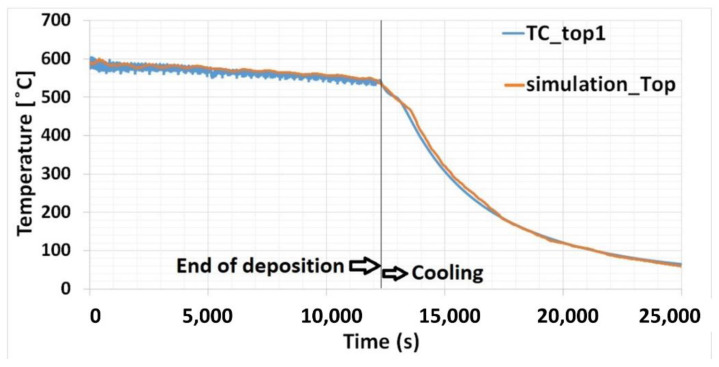
Temperature distribution on the baseplate top surface—experimental and numerical results.

**Figure 7 materials-15-04165-f007:**
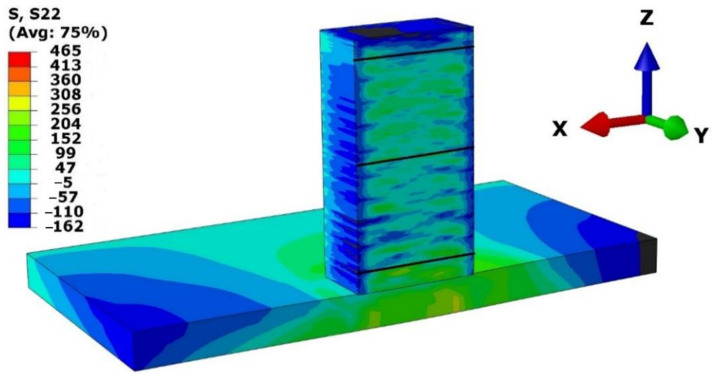
Normal stress distribution at the cut surface of X-Z plane in MPa.

**Figure 8 materials-15-04165-f008:**
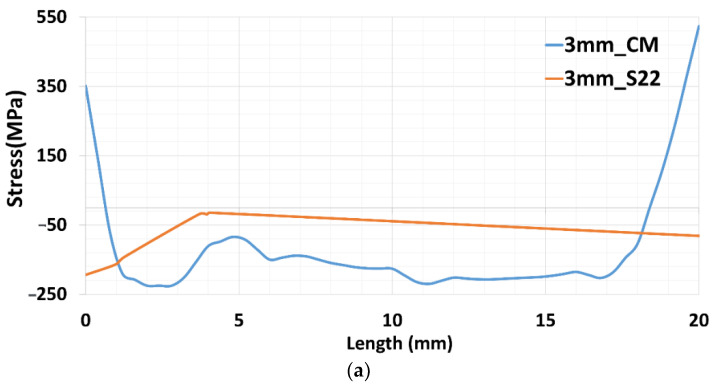
Residual stress in the three paths for contour method and numerical results. (**a**) path at a distance of 3 mm. (**b**) path at 18.75 mm. (**c**) path at 35.5 mm from the top surface of the baseplate.

**Figure 9 materials-15-04165-f009:**
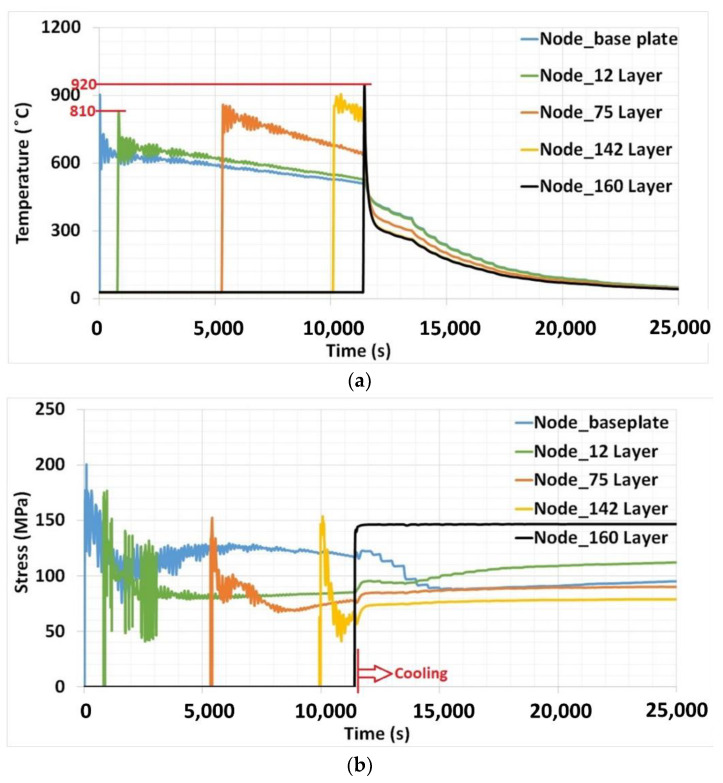
Nodal results at different layers in the deposited structure: (**a**) temperature, (**b**) thermal stress.

**Figure 10 materials-15-04165-f010:**
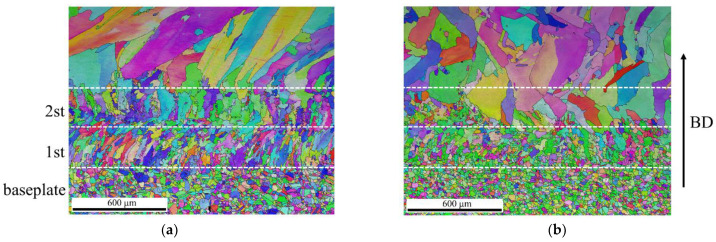
Comparison of microstructure in the vicinity of the baseplate for samples produced (**a**) without preheating (WPH), and (**b**) with elevated temperature preheating of 600 °C (EPH).

**Figure 11 materials-15-04165-f011:**
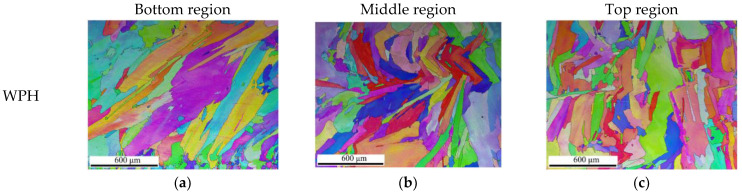
EBSD Inverse Pole Figures (IPF) showing the difference in grain structure at different regions: (**a**–**c**) WPH samples, (**d**–**f**) EPH samples.

**Figure 12 materials-15-04165-f012:**
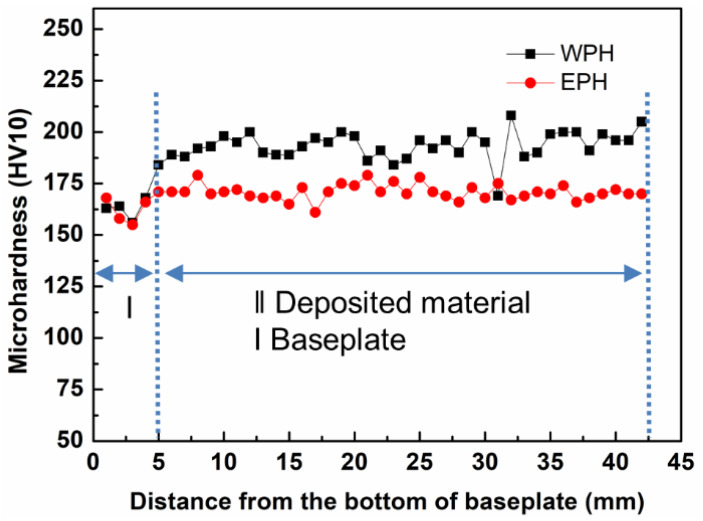
Hardness dependence of the distance from the bottom of the baseplate (X-Y plane).

**Figure 13 materials-15-04165-f013:**
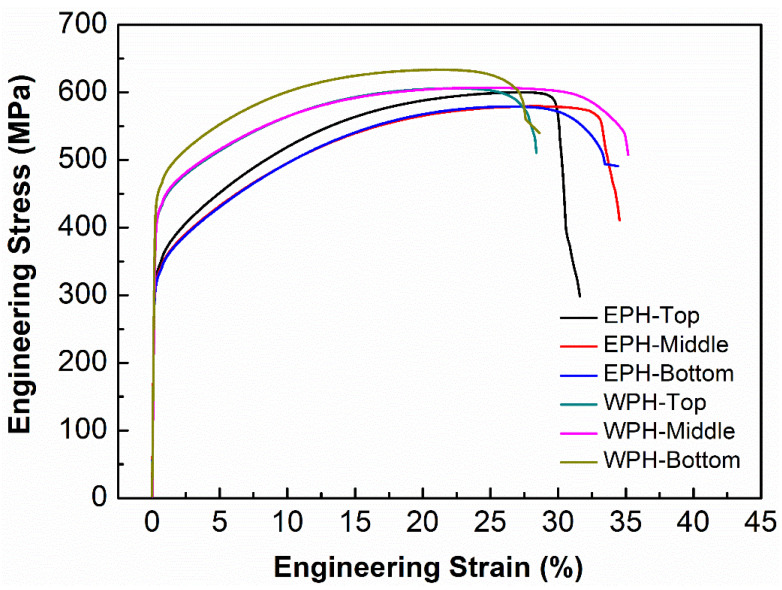
Engineering stress–strain curves of WPH and EPH samples at different locations.

**Figure 14 materials-15-04165-f014:**
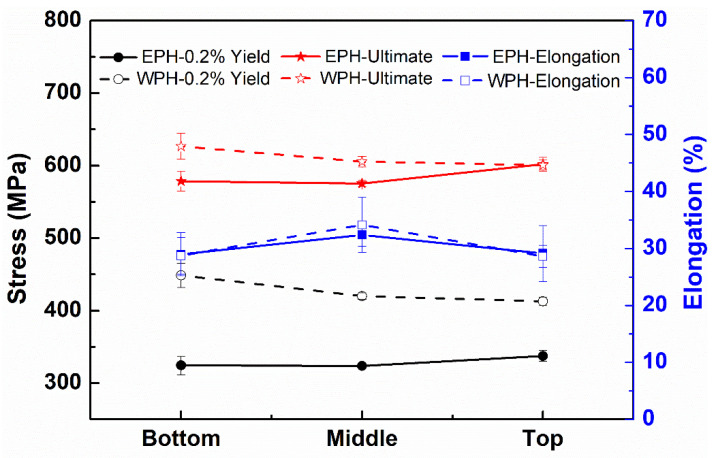
Dependence of 0.2% yield strength, ultimate strength, and elongation of WPH and EPH specimens on location.

**Table 1 materials-15-04165-t001:** Summary of the average tensile properties of DED-SS316L specimens.

	0.2% Yield Strength (MPa)			Ultimate Strength (MPa)			Elongation (%)		
Location	WPH	EPH	Decrease by	WPH	EPH	Decrease by	WPH	EPH	Decrease by
Bottom	449	324	28%	627	579	8%	29	29	−1%
Middle	420	324	23%	605	575	5%	34	32	5%
Top	413	338	18%	601	602	−1%	28	29	−1.75%

## Data Availability

Data available in a publicly accessible repository.

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
