# Peer review of "Elevated Temperature Baseplate Effect on Microstructure, Mechanical Properties, and Thermal Stress Evaluation by Numerical Simulation for Austenite Stainless Steel 316L Fabricated by Directed Energy Deposition"

_materials, 2022, doi:10.3390/ma15124165_

Round 1

Reviewer 1 Report

This paper investigates the effect of a continuous heating for the baseplate during direct energy deposition on the final thermal stress, microstructure, and mechanical properties using austenitic steel 316L. The investigation of thermal stress was conducted by numerical simulation and compared with experimental results. All experiments are carefully designed. Both numerical model and experimental procedures are well explained and the results are properly present. This paper is acceptable after minor revision:

  1. Line 55: does preheating reduce residual stresses of 80.2% or by 80.2%? please double check.
  2. Line 79: reports → report
  3. Line 109: chemical properties → chemical compositions
  4. Line 109: cross reference Table 1, for example: add “in Table 1” at the end of the sentence
  5. Line 112: delete the whole sentence
  6. Figure 9: suggest to present the results of deposition on unheated baseplate instead of just citing the reference in text, which could help reads greatly to have a better understanding of the discussion in the text. If they were published before, remember to request figure permissions.

Author Response

Thank the reviewer for those valuable suggestions.

  1. we checked and modified the sentence: ‘reduce residual stresses and final distortion by 80.2% and 90.1%, respectively’ in line 55.
  2. Lines 80-89 were revised, and we would like to draw you attention to this new part.
  3. Yes, we edited. As this information was given in previous paper, we deleted this repeated information, please check in line ll2-115.
  4. Lines 112-115 were rewrote and please have a look in the new version.
  5. Yes, we deleted.
  6. Yes, we also presented the results of deposition on unheated baseplate for better comparison. Please have a look at the new Figure 8 (the number of figure was changed as we edited all figures according to other reviewers.)

Reviewer 2 Report

The paper presents the effect of baseplate preheating on the material deposition process as compared to the case of a baseplate at room temperature.

The researches presented in the paper are thorough, featuring both an experimental part and a numerical simulation part.

However, while even the start of the conclusions reiterates the idea of a comparison, the researches themselves and the conclusions are rather inconsistent in this regard. Maybe the comparison aspect needs to be reinforced to some extent. 

In line 140, the term ”demission” should probably be “dimension”.

Author Response

Thank the reviewer for the valuable comments.

  1. According to the suggestions, we revised conclusions to make all results more consistent. please check the changes in conclusions in the new version.
  2. Yes, we corrected the word, it should be 'dimension'.

Reviewer 3 Report

Dear Authors;

The paper focuses on the effect of temperature on microstructure, mechanical properties, and thermal stress evaluation by numerical simulation for austenite steel 316L prepared via directed energy. The manuscript is well written and well organized. The work is not original.

I suggest that the authors should pay much attention to revise the manuscript by taking following points into account:

- Abstract must be enriched via valuable results, which pave the way for understanding the audiences.

- The authors should more clearly emphasis the novelty of their work in the introduction.

- Authors need to elaborate introduction and rephrase it because it contain plagiarism.

- Line 103: “2.1 Material” → “2.1 Materials”.

- Authors need to delete Figure 1 and Table 1 because they have already been mentioned in the published work (Materials 2020, 13(11), 2666; https://doi.org/10.3390/ma13112666).

- Line 125: Table 2. DED process parameters. The authors should provide the corresponding reference.

- Authors need to delete “Figure 3. Thermocouple position and structure dimension” because it has already been published.

- 2.6.1 Thermal analysis: it is useless to rewrite the equations because they have already been published. Authors can put the previous publication as a reference.

- L 275-L276: “it is likely due to 275 error occurred during cutting process”. The authors should provide the corresponding reference.

- Authors need to pay attention in Reference section also for formatting. Some of the ref. ([15], [19], [31], [32], and [33]) have missing page no and/or volume.

Sincerely Yours

Author Response

Thank the reviewer for all the valuable suggestions and constructive comments which we believe can help us make this work better and easier understandable.

  1. We modified the abstract according to the suggestions. Please check the abstract in the new version.
  2. The novelty of the present work was highlighed in the revised version. We would like to draw your attention to lines 80-100.
  3. The introduction was edited in the new manuscript.we would like to draw your attention to highlighed parts in the introduction in the new version.
  4. Modified
  5. Figure 1 and Table 1 was deleted.
  6. The corresponding reference was added, please check in line 120 in the new revised version.
  7. Thank the reviewer for pointing out this question. But we think it is better to keep this figure. In our previous paper, two thermocouples were used and depicted, whereas, in the present work, only thermocouple were placed on the top of baseplate. Presenting the position of the thermocouple can help reader better understand the experimental design.
  8. We deleted equations and cite previous paper as suggested, please check thermal analysis part in the new version.
  9. The corresponding references were given in the sentence following “it is likely due to” which explained the reason for the error of contour results. We would like to draw your attention on “cutting edge influences residual stress relaxation [20,31].”
  10. Missing page and/or volume of those references were added and other references were also checked.

Reviewer 4 Report

1.        The title is incorrect. It should specifically indicate that the material is austenite stainless steel 316L to instead of austenite steel 316L.

2.        Abstract is improper and wordy. Abstract must be straight to the point.

3.        In the keyword, microstructure and mechanical property should be removed because both are too general.

4.        On the page 3, the paragraph to introduce the aim of this study should be at the last paragraph after you explain all the literature review in introduction.

5.        On the page 4, the author should introduce what kind of test to obtain 50-150 micrometer.

6.        On the page 6, to indicate where the thermocouple position is since it is important for measuring in the figure.

7.        On the page 7, does the test use the standardization of tensile test specimens such as ISO 6892-1? If yes, just present it. If no, mention the standard. If it is redesigned, the size and dimension should be described.

8.        On the page 8, add the photo with indentation profile of hardness test.

9.        On the page 13, Figure 6(b) was not presented.

10.     On the page 14, Figure 7(b) was not presented.

11.     One the page 16, figures and explanations must be in sync. Do not forget to mention and explain each figure you added in the manuscript.

12.     On the page 17, how to obtain the Figure 8?

13.     On the page 18, the reviewer cannot catch which figure that the author tries to explain in the second and third paragraphs.

14.     On the page 19, using hardness test to instead of vickers hardness. Author should explain the relationship between hardness and microstructure.

15.     On the page 20, why WPH has highest strength and lowest strain? give some explanation about tensile relationship with microstructure.

16.     On the page 22, give a comparison instead of mentioning the numbers already in the table.

17.     In the conclusion, point two should be noticed that preheat can change the microstructure to be fine structure especially for the first layer.

18.     In the conclusion, point three should be noticed that preheat can increase the hardness and tensile strength of material, but no significant effect for strain.

Author Response

Thank the reviewer sincerely for the valuable and constructive comments, which have certainly helped to improve the presentation and quality of our paper. We have updated our paper according to the suggestions as requested by the reviewer.

Note that all modification was highlighted in red in the new version as well as using Track of changes requested by the journal. As we noticed there was a page issue which may due to the different versions of MS Word/LaTeX. The same content mentioned in the comment below was in different pages here. Therefore, for accuracy, all revision in the new manuscript and the answers will be mentioned more clearly, such as pages, section, lines. Using track of changes in All Mark-up mode can be a little messy and also affect the line number since substantial changes were made in the last two round reviewing for this manuscript. And we still keep all changes for this manuscript. We suggest reading the new version in the Simple Mark-up mode and all pages, lines answered below were recorded in Simple Mark-up mode. Don’t worry, the modification was also highlight in the new version and can be clearly seen.

  1. Corrected in the new version!
  2. Abstract was modified in the revised manuscript according to the suggestion.
  3. Microstructure was removed and mechanical property was replaced with tensile properties in the new version.
  4. The aim of this study was placed in the last paragraph in the introduction. Some details of experimental design different from previous work was also depicted in the last paragraph, as you can see lines 84-88 in new version.
  5. The information of powder size was provided by manufacturer with a certificate showing powder size of 50-150 micrometer.
  6. We agree with the reviewer. The schematic of thermocouple position was added in the new version. please have a look at new figure 2.The schematic of thermocouple position was given in our previous version, but the previous reviewer 3 insisted to delete the figure as it was depicted in our previous work [20].
  7. Tensile test specimen used in the present study was not the standard tensile specimen as ISO 6892. They are miniaturized tensile sample according to ISO 6892-1 in order to save AM-materials. previous study about the small tensile sample can be found in [21.22]. The size and dimension can be seen in Figure 3(d).
  8. The figure of indentation profile of hardness test was added in the new figure 4 in the modified version.
  9. Figure 6(b) and figure 7(b) below can be seen under Simple Markup mode. The old figure 6(b) is the new figure 8(b), as we added some figures. We figured out there may be format problem if the manuscript was open in the All Markup mode due to the version of the word. In case of misunderstanding, please try to read the new manuscript using both All Markup and simple Markup.
  10. Figure 7(b) can be seen in the new manuscript using Simple Markup mode. The new number is Figure 9(b).
  11. All figures and the corresponding explanations in the text was checked and edited carefully. We would like to draw your attention to the red highlighted in page 11-12 in the new version.
  12. Figure 8 was EBSD maps obtained by choosing area where deposited layers near the baseplate. That is why we can see an evident transition between grain orientations, grain size and grain morphology. This figure was given in order to clearly demonstrate the effect of baseplate heating on the first several deposited layers near the baseplate
  13. We are sorry for this inconvenience. The figure and corresponding text was also checked and specific Figure was inserted in the proper position in the new version. Please have a look at the highlighted in red in the page 13, line 346. 347, 361-363.
  14. Corrected. The relationship between hardness and microstructure was added in the page 14, section 3.5.1 hardness, line 382-384, line 388-390.
  15. Explanations about tensile result and microstructure was given in page 14, section 3.5.2 tensile properties, line 397--398.
  16. Comparison regarding the elevated temperature baseplate preheating effect range was given in the new version, in page 16, lines 420-424.
  17. Thanks for this useful notice. Point two was revised according to the suggestion. Please have a look at page 17, line 455-456, 459-461.
  18. Thanks for pointing out this results. We added this important result in the point three, as can be seen in page 17, line 462-465.

Round 2

Reviewer 3 Report

Dear Authors;

Thanks for your work in revising your manuscript according to the suggested

comments. The revised paper is well improved. 

I suggest that the authors delete all Figures and/or Tables that have already

been published.  

I hope that the revised manuscript is now acceptable for publication.

Gud luck and Best Regards 

Author Response

We sincerely thank the reviewer for pointing out the valuable question.

We deleted all figures or tables that have already been published.

For example, old Table 1 DED process parameters and cited previous work;

Old Figure 2 which demonstrates the thermocouple position, and cite the previous work.

Old Figure 9 which adds the thermal results of room temperature baseplate condition according to the suggestion of another reviewer., and only keep the thermal results for elevated temperature baseplate condition.

All number of figures and tables were changed.

We hope all changes can address the reviewer concern. And we sincerely thank the reviewer all valuable comments.

Best regards,